# TMT-Based Quantitative Proteomics Analysis Reveals Airborne PM_2.5_-Induced Pulmonary Fibrosis

**DOI:** 10.3390/ijerph16010098

**Published:** 2018-12-31

**Authors:** Shan Liu, Wei Zhang, Fang Zhang, Peter Roepstorff, Fuquan Yang, Zhongbing Lu, Wenjun Ding

**Affiliations:** 1Laboratory of Environment and Health, College of Life Sciences, University of Chinese Academy of Sciences, Beijing 100049, China; liushan14@mails.ucas.ac.cn (S.L.); zhangw@ucas.ac.cn (W.Z.); luzhongbing@ucas.ac.cn (Z.L.); 2Sino-Danish College, University of Chinese Academy of Sciences, Beijing 100190, China; 3Department of Biochemistry and Molecular Biology, University of Southern Denmark, DK-5230 Odense M, Denmark; roe@bmb.sdu.dk; 4National Laboratory of Biomacromolecules, Institute of Biophysics, Chinese Academy of Sciences, Beijing 100101, China; fqyang@ibp.ac.cn

**Keywords:** particulate matter (PM_2.5_), pulmonary fibrosis, toxicity, quantitative proteomics

## Abstract

Epidemiological and experimental studies have documented that long-term exposure to fine particulate matter (PM_2.5_) increases the risk of respiratory diseases. However, the details of the underlying mechanism remain unclear. In this study, male C57BL/6 mice were exposed to ambient PM_2.5_ (mean daily concentration ~64 µg/m^3^) for 12 weeks through a “real-world” airborne PM_2.5_ exposure system. We found that PM_2.5_ caused severe lung injury in mice as evidenced by histopathological examination. Then, tandem mass tag (TMT) labeling quantitative proteomic technology was performed to analyze protein expression profiling in the lungs from control and PM_2.5_-exposed mice. A total of 32 proteins were differentially expressed in PM_2.5_-exposed lungs versus the controls. Among these proteins, 24 and 8 proteins were up- and down-regulated, respectively. Gene ontology analysis indicated that PM_2.5_ exerts a toxic effect on lungs by affecting multiple biological processes, including oxidoreductase activity, receptor activity, and protein binding. Furthermore, Kyoto Encyclopedia of Genes and Genomes (KEGG) analysis revealed that extracellular matrix (ECM)–receptor interaction, phagosome, small cell lung cancer, and phosphatidylinositol 3-kinase(PI3K)-protein kinase B (Akt) signaling pathways contribute to PM_2.5_-induced pulmonary fibrosis. Taken together, these results provide a comprehensive proteomics analysis to further understanding of the molecular mechanisms underlying PM_2.5_-elicited pulmonary disease.

## 1. Introduction

Numerous epidemiological and clinical studies have demonstrated that long-term exposure to higher concentrations of ambient airborne particulate matter (PM) with a mean aerodynamic diameter of <2.5 μm (PM_2.5_) is associated with various health outcomes, including mortality, hospitalization for respiratory and cardiovascular diseases, aggravation of asthma attacks, and lung cancer [1,2,3,4]. Animal experiments have also documented that PM_2.5_ triggers pulmonary inflammation, oxidative stress, and worsened lung impedance and histology in mice [5,6,7], which may result in lung fibrosis [2]. However, the molecular mechanism underlying PM_2.5_-induced lung injury has yet to be fully elucidated.

Pulmonary fibrosis is characterized by the deposition of collagen and other extracellular matrix molecules with persistence of fibroblasts/myofibroblasts [3]. Previous studies have suggested that oxidative stress caused by reactive oxygen species (ROS) overproduction may be directly or indirectly involved in the pathogenesis of human pulmonary fibrosis [4]. It is well established that PM_2.5_ exposure triggers an increase of ROS in human lung alveolar epithelial cells and idiopathic pulmonary fibrosis patients [8,9,10]. The chemical composition and oxidative potential play crucial roles in PM_2.5_-induced toxicity [11]. Importantly, oxidative stress can further activate redox-sensitive signaling cascades which result in inflammation response [6,12]. Pro- and anti-inflammatory cytokines and growth factors as well as non-collagenous extracellular matrix proteins including tumor necrosis factor (TNF-α) [13], transforming growth factor (TGF-β) [14], and matrix metalloproteinase (MMP) [15] have been implicated in PM_2.5_-induced alveolar oxidative damage.

Mass spectrometry (MS)-based proteomics in combination with complementary analytical techniques are enabling novel insights into the modulation of particle surfaces by biological fluids and subsequent particle-induced cellular responses [16,17,18]. Tandem mass tags (TMT) quantitative mass spectrometry combined with the multidimensional protein identification technology (MudPIT) technique enable the genome-wide quantification of protein expression levels under adaptive responses to environmental, physiological, or pathological conditions [19]. A recent study reported the cytotoxicity of water-soluble PM_2.5_ extract exposure on human lung epithelial cells (A549) at the proteomic level and found an array of differential proteins involved in oxidative stress response to PM_2.5_ [20]. In this study, we exposed C57BL/6 mice to either PM_2.5_ or filtered air (FA) for 12 weeks through a whole-body PM_2.5_ exposure system and then examined the proteins associated with PM_2.5_-induced fibrosis using TMT combined with a liquid chromatography–tandem mass spectrometry (LC-MS/MS) proteomics technique. The proteins that are potential biomarkers for lung injury induced by PM_2.5_ were particularly characterized.

## 2. Materials and Methods 

### 2.1. Reagents

Hematoxylin and eosin (H&E) staining solution and Sirius red staining solution were purchased from Leagene Biotech Co., Ltd. (Beijing, China). Trypsin was purchased from Promega (Madison, WI, USA). Tandem mass tags 6-plex reagents were purchased from Thermo Fisher Scientific (Waltham, MA, USA). Triethylamine borane (TEAB), sodium dodecyl sulfate (SDS), DL-dithiothreitol (DTT), and urea were obtained from Sigma (St. Louis, MO, USA). The RNA simple Total RNA Kit and lysis buffer were purchased from Tiangen Biotech Co., Ltd. (Beijing, China). The GoScript Reverse Transcription System and GoTaq qPCR Master Mix were purchased from Promega Corporation (Madison, WI, USA).

### 2.2. Animals and Whole-Body Inhalation

Five-week-old male C57BL/6 mice were obtained from the experimental animal care center of the First Affiliated Hospital of the People’s Liberation Army (PLA) General Hospital. The animals were maintained at 24 °C on a 12 h light/12 h dark cycle. They had free access to standard laboratory chow and water. The experimental protocols and the use of animals were approved by the Experimental Animal Centre, the First Hospital Affiliated to the Chinese PLA General Hospital. The animals were cared for in accordance with the principles of the Guide for Care and Use of Experimental Animals. All mice were cared for in accordance with ethical guidelines set forth by the Experimental Animal Centre, the First Hospital Affiliated to the Chinese PLA General Hospital, with the Institutional Animal Care and Use Committee (IACUC; #SYXK 201-0014).

As previously described by Wan et al. [7], all mice were exposed by inhalation to either filtered air (FA) or PM_2.5_ for 12 h/day, 5 days/week from May to August 2015 (a total duration of 82 days; about 12 weeks). Inhalation exposure was carried out in a “real-world” airborne PM_2.5_ exposure system in ZhongGuancCun Campus of University of Chinese Academy of Sciences (N 39°57′39.83″ E 116°20′10.97″). The animal groups were randomly divided into two groups: FA (*n* = 6) and PM (*n* = 6). During the whole exposure stage, body weight and food and water consumption were monitored daily.

### 2.3. PM_2.5_ Sampling and Physical and Chemical Characterization

To calculate the PM_2.5_ mass concentrations in the exposure chambers, daily ambient PM_2.5_ samples were collected continually during the exposure time period. The sampling site was chosen on a rooftop (about 30 m above ground) in ZhongGuanCun Campus of University of Chinese Academy of Sciences (UCAS), surrounded by some institutes and residential areas. There is high traffic flow and a high population density in the daytime. Large industrial and thermoelectric plants are absent from the area. The distance of the sampling inlets from the main road was 50 m. PM_2.5_ samples were collected on Teflon filters (diameter = 47 mm; Whatman, Piscataway, NJ, USA) for biological assay using a low-volume sampler (42 L/min, URG, Chapel Hill, NC, USA) for 12 h (8:00–20:00). Before and after the sampling, the Teflon filters were equilibrated in conditions of 30% relative humidity and 25 °C room temperature for over 48 h and then weighed on a high-precision microbalance (Mettler Toledo, OH, USA) to measure the collected atmospheric daily PM_2.5_ concentration. All sampled filters were stored in darkness at −20 °C before further chemical and physical characterization.

The PM_2.5_ samples on the Teflon filters were prepared according to the methods exactly as we described previously [21,22]. Briefly, PM_2.5_ samples were extracted from the sampled filters by immersing them in deionized water (18.2 MΩ/cm) and sonicating for 30 min in a water bath sonicator (KQ-700 V, 700 W). The extracted samples were stored at −80 °C prior to analysis. The physical and chemical characterization of the sample extracts was then conducted as described previously [21,22]. In brief, the size distribution of PM_2.5_ was measured using scanning electron microscopy (SEM, JSM-6700 F, JEOL, Tokoy, Japan) at a magnification of 5000×–10,000× and accelerating voltage of 5 kV. The size distribution of PM_2.5_ in suspension was analyzed using a Nano-Zetasizer (1000 HS; Malvern Instrument Ltd., Worcestershire, UK). The inorganic elements of the collected PM_2.5_ were detected by acid digestion (HNO_3_/HF = 7:3), followed by measurement using inductively coupled plasma mass spectrometry (ICP-MS, Thermo, Elemental X7, Waltham, MA, USA). Organic and elemental carbon (EC) were measured on-filter using a thermal–optical analyzer (Sunset Laboratories, Hillsborough, NC, USA). The water-soluble inorganic components (SO_4_^2−^, NO_3_^−^, NH_4_^+^, and Cl^−^) were determined using ion chromatography (Dionex-600, Sunnyvale, CA, USA).

### 2.4. Lung Preparation and Histopathological Examination

After 12 weeks of exposure, the mice were anesthetized with ether. Blood samples were collected from the abdominal vein with a microsyringe. Serum was separated at 3000 rpm for 15 min. The lung of each mouse was immediately excised. After washing with saline, the right lungs were frozen in liquid nitrogen until proteomic analysis. Segments of left lung were fixed with 4% neutral buffered formalin immediately and embedded in paraffin. Tissues were embedded in paraffin at 60 °C and sectioned at 5 μm thickness using an automatic microtome. Paraffinized lung sections were deparaffinized and stained with H&E or Sirius Red to observe tissue morphology. More than five prepared histological lung section samples per tissue per group were observed with an optical microscope (Leica DM4000, Germany). ImageJ (version 1.51) was used to perform semi-quantification of the Sirius red slide images.

### 2.5. Protein Extraction and Digestion

Protein preparation from the lungs of mice was performed according to a method previously described [23] with some modifications. Briefly, lung tissue (30 mg) from three mice was pooled and homogenized by a Teflon homogenizer in 5 volumes (*v*/*w*) of isolation buffer consisting of 4% SDS and 0.1 M Tris-HCl, 10 mM DTT, 8 M urea and 1% protease inhibitor cocktail (pH = 8.0, obtained from Sigma). After 1 min of homogenization on ice, the samples were sonicated using a Scientz-IID sonicator (Scientz, China) for 10 min on an ice–water mixture. The homogenate was centrifuged at 4 °C and 20,000*g* for 10 min. The supernatant was collected and the total protein concentration was determined using bicinchoninic acid protein assay kits. A quantity of 200 μg of proteins was reduced by 10 mM DTT at 37 °C for at least 1 h followed with 20 mM iodoacetamide (IAM) for 30 min in the dark. The protein solution was diluted 1:5 with 50 mM NH_4_HCO_3_ and digested with 0.5 μg/μL trypsin at 37 °C overnight, followed by stopping the reaction with formic acid. The digestion was desalted on hydrophilic-lipophilic balanced (HLB) C_18_ cartridge columns (Waters, MA, USA), and peptides eluted with 60% acetonitrile were lyophilized via vacuum centrifugation.

### 2.6. Tandem Mass Tags Labeling and High-Performance Liquid Chromatography Fractionation

According to the manufacturer’s protocol for the TMT Kit, equal amounts of desalted peptide (50 μg) from each pooled sample were reconstituted in 100 mM Triethylamine borane and labeled with TMT 6-plex reagent (127 for the FA group and 129 for the PM group). Peptides were combined 1:1 and lyophilized. 

Before LC-MS/MS analysis, the TMT-labelled samples (100 μL) were prefractionated on a XBridge BEH130 C_18_ column (Rigol high-performance liquid chromatography (HPLC) system, 250 mm × 4.6 mm; particle size: 5 μm) to reduce complexity. The mobile phases were as follows: Solvent A, 2% acetonitrile, pH 10; and Solvent B, 98% acetonitrile with 2% H_2_O, pH 10. After equilibration, a linear gradient was started as follows: 0–5 min, 95% A, 5% B; 5–40 min, 92% A, 8% B; 40–62 min, 82% A, 18% B; 62–64 min: 68% A, 32% B; 64–68 min: 5% A, 95% B; 68–69 min: 5% A, 95% B; 69–76 min: 95% A, 5% B. A 0.7 mL/min velocity was used throughout the whole process. A total of 38 fractions were collected and combined into 13 fractions by combining 1, 13, and 25; 2, 14, and 26; 3, 15, and 27; and so on. The last tube combined 37 and 38. The combined fractions were dried, lyophilized, and stored at −80 °C until LC-MS/MS analysis.

### 2.7. Reversed-Phase Liquid Chromatography–Tandem Mass Spectrometry

The NanoLC-MS/MS experiments were performed using an EASY-nLC1000 HPLC system (Thermo Fisher) interfaced with a Q Exactive (Thermo Fisher). The labeled peptides were resolved in 20 μL 0.1% formic acid, and then loaded onto a 100 μm inner diameter (i.d.) × 2 cm fused silica trap column packed in-house with reversed-phase silica (Reprosil-Pur C18 AQ, 5 μm, Dr. Maisch GmbH, Tübingen, Germany). The peptides were eluted with a 78 min linear gradient: 5–8% solvent B (90% ACN, 0.1% FA), 8 min; 8–22% solvent B, 50 min; 22–32% solvent B, 12 min; 32–95% solvent B, 1 min at a flow rate of 280 nL/min. The mass spectrometer was operated in data-dependent mode (DDA). The MS1 survey scan was from 300–1600 *m*/*z*, and data were acquired at a high resolution of 70,000 (*m*/*z* 200). The target value was 3 × 10^6^ with a maximum ion injection time of 60 ms. As for the second stage of mass spectrometry (MS2) scans, the top 20 precursor ions were selected from the first stage of mass spectrometry (MS1) full scan with an isolation width of 2 *m*/*z* from fragmentation in high-energy collision dissociation (HCD; normalized collision energy = 32%). The MS2 spectra were acquired at a resolution of 17,500 (*m*/*z* 200). The dynamic exclusion time was 40 s. The target value was 5 × 10^4^ with a maximum ion injection time of 80 ms. For the setting of the nano electrospray ion source, the spray voltage was 2.0 kV; no sheath gas flow; and the heated capillary temperature was 320 °C.

### 2.8. Data Processing and Database Searching

The MS/MS spectra were processed using Proteome Discover version 2.2 (Thermo Fisher), and database searches were carried out against a target and decoy separated *Mus musculus* database downloaded from Swiss-Prot (Sprot 2015_10 version; 70,566 sequences) using an in-house Mascot server (version 2.3.0; Matrix Science Ltd., London, UK). Trypsin was chosen as the enzyme, allowing up to two missed cleavage sites. The fragment mass tolerance for MS/MS spectra was set to 0.02 Da. Carbamidomethylation was chosen as a static modification. Oxidation and TMT 6-plex (lysine and N-terminus of peptides) were specified as dynamic modifications. Only rank 1 peptides and a false discovery rate (FDR) of ≤1% were accepted in this study. Ratios for each TMT label were obtained using a wild-type mouse sample as the denominator.

### 2.9. Data Normalization and Protein Network Analysis

Proteins found in three replications and with more than two peptides were selected for further analysis. Significantly regulated proteins between experimental groups were determined based on their *p*-value. Only proteins with more than 1.30-fold or less than 0.769-fold change compared to control groups and with standard deviation less than 2 times the median were considered differentially regulated.

Gene ontology (GO) was performed using PANTHER version 12.0 (http://www.pantherdb.org/), and DAVID Bioinformatics Resources 6.8. Kyoto encyclopedia of genes and genomes (KEGG) pathway analysis was performed using DAVID. Student’s *t*-test with a *p*-value of <0.05 was considered significant. The STRING (Search Tool for the Retrieval of Interacting Genes/Proteins) algorithm was used to build protein–protein interaction networks. Only high-confidence interactions (score ≥ 0.7) were chosen.

### 2.10. Quantitative Real-Time Polymerase Chain Reaction Analyses

The total RNA of the lung tissue was extracted using total RNA pure kits (Tiangen, Beijing, China) according to the manufacturer’s instructions. The concentrations of RNA were measured using a Nanodrop 2000 (Thermo Fisher). First-stand cDNA was synthesized from total RNA using a first-stand cDNA synthesis kit (Promega, Madison, WI, USA). Quantitative real-time polymerase chain reaction (qPCR) was performed using SYBR green supermix (Promega, Madison, WI, USA) and PCR primers in Biomed (Beijing, China). The primer sequences for target genes are listed in Appendix A. The amplification was performed in a total mixture volume of 10 μL. The PCR cycle was as follows: 1 cycle at 95 °C for 5 min, then 40 cycles at 95 °C for 30 s, at 55 °C for 1 min and at 72 °C for 45 s. β-actin mRNA was used to normalize variations in amplification efficiency. The results are presented as the mean ± standard deviation (SD) of three independent experiments. Significant differences between groups were evaluated by Student’s *t*-test. In all cases, a *p*-value less than 0.05 was considered to present a statistically significant difference.

## 3. Results

### 3.1. Physiochemical Characterization of PM_2.5_

During the period of PM_2.5_ exposure, the mean daily ambient PM_2.5_ concentration at the study site was about 64 μg/m^3^ (the annual average PM_2.5_ National Ambient Air Quality Standard (NAAQS) is 15 μg/m^3^ in China). The morphology of PM_2.5_ particles found by SEM is shown in Figure 1A. In addition, the data from the dynamic light scattering measurement showed a size range of 0.6–0.8 μm with a mean of 0.7 μm (Figure 1B). 

Among the total 21 metal elements measured in the PM_2.5_ samples, the most abundant elements were Ca (13.76 ± 0.30 μg/mg), K (10.33 ± 0.03 μg/mg), Na (7.96 ± 0.08 μg/mg), Fe (7.056 ± 0.13 μg/mg), and Al (7.06 ± 0.13 μg/mg), and the content of heavy metals such as Zn, Pb, Mn, Cu, and V was also very high (Table 1). The analytical results of SO_4_^2−^, NO_3_^−^, NH_4_^+^, and Cl^−^ levels in the PM_2.5_ were 212.71 ± 3.67, 97.52 ± 6.81, 85.61 ± 0.52, and 1.85 ± 0.41 μg/mg, respectively (Table 1). The average concentrations of organic carbon (OC) and elemental carbon (EC) were 44.09 ± 0.32 and 5.51 ± 0.75 μg/mg, and the OC/EC ratio was 1.72 μg/mg (Table 1). 

### 3.2. PM_2.5_ Exposure Induces Pulmonary Inflammation and Fibrosis

As shown in Figure 2A, histological lung sections obtained from PM_2.5_-exposed mice exhibited striking pathological alterations of pulmonary inflammation and fibrosis compared to the FA groups. The results of qPCR showed that PM_2.5_ exposure significantly up-regulated the expression levels of collagen I and transforming growth factor beta (TGF-β) mRNA in the lungs (Figure 2B,C). Moreover, PM_2.5_ exposure increased serum tumor necrosis factor α (TNF-α) and interleukin 6 (IL-6) (Figure 2D,E).

### 3.3. Quantitative Proteomics Analyses to Reveal PM_2.5_-Regulated Pulmonary Proteins

To analyze the expression changes of proteins associated with PM_2.5_-induced pulmonary injury and fibrosis, a quantitative proteomic analysis was carried out to characterize the protein samples obtained from the lungs of FA- and PM_2.5_-exposed mice using TMT labeling and LC-MS/MS techniques. In total, we identified and quantified 7384 proteins (Appendix A), of which 6581 were proteins with at least 2 unique peptides. Based on Student’s *t*-test with *p* ≤ 0.05 and 1.3-fold change, 32 of the 6581 quantified proteins were considered to be significantly differently accumulated. Among these proteins, 24 and 8 proteins were up- and down-regulated, respectively (Appendix A).

To understand an overview of the effects of PM_2.5_ exposure on lungs, the differential proteins were categorized according to their functional properties by searching the GO database (http://www.pantherdb.org/). As shown in Figure 3, in accordance with the cellular compartments and molecular functions of the GO term, the largest proportion of proteins (15%) was associated with oxidoreductase activity and receptor activity, and other proteins were mainly involved in protein binding, signal transduction, transport, antioxidant activity, biological adhesion, and other functions. Therefore, our results suggested that PM_2.5_ exposure disrupts the pulmonary homeostasis, leading to the activation of cellular mechanisms to overcome PM_2.5_-induced oxidative stress.

### 3.4. Biological Interaction of Differentially Expressed Proteins in Response to PM_2.5_ Exposure

To further determine the biological interaction of the observed 32 differentially expressed proteins in response to PM_2.5_ exposure, KEGG pathway analysis was carried out. The results of the KEGG pathway analysis showed that the differentially expressed proteins were associated with 7 pathways (*p* < 0.05). As shown in Table 2, the most enriched pathway was in cancer, followed by extracellular matrix (ECM)–receptor interaction, phagosomes, the PI3K–Akt signaling pathway, small cell lung cancer, protein digestion and absorption, and amoebiasis. Moreover, we found that fibrosis-related proteins, such as collagen alpha-4 proteins, fibroblast growth factor 1, matrix metalloproteinase-9, cytochrome b-245, and CD36, were significantly up-regulated after PM_2.5_ exposure (Table 2). Additional differentially expressed proteins consisted of a set of redox homeostasis regulatory proteins such as ubiquinone biosynthesis monooxygenase, myeloperoxidase, cytochrome b-245 heavy chain, cytochrome b, and cytochrome p450; the immune-related proteins were also found, such as histone H2B type 2-E, platelet glycoprotein 4, disintegrin and metalloproteinase domain-containing protein 17, collagen alpha-2(IV) chain, collagen alpha-1(IV) chain, and BPI fold-containing family B member 1 (Table 3). In addition, the network analysis of protein–protein interaction showed that PM_2.5_ exposure induced ECM–receptor interaction, phagosomes, and pathways in cancer (Figure 4). Taken together, these results suggest that these proteins are involved in redox homeostasis and in immune and inflammatory responses which may contribute to PM_2.5_-induced pulmonary injury and fibrosis.

### 3.5. Verification of Protein Expression by RT-PCR Analysis

To validate the differential expression of lung-fibrosis-related proteins identified and quantified after PM_2.5_ exposure, we further performed RT-PCR analysis to identify the values of the expression of the nine selected proteins (Figure 5). The levels of mRNA were similar to those of the proteomic data.

## 4. Discussion

The present study demonstrates that PM_2.5_ exposure led to inflammatory and fibrotic responses in the lungs of mice. In this study, we further performed comparative quantitative proteomics analyses for PM_2.5_-induced lung injury. Consequently, 32 differentially expressed proteins were identified and quantified. These proteins include redox homeostasis, ECM–receptor interaction, phagosomes, the PI3K–Akt signaling pathway, small cell lung cancer, and fibrosis. Taken together, our results indicate that multiple pathways were activated by PM_2.5_-induced oxidative stress in lung injury.

### 4.1. Characterization of PM_2.5_

Fine and ultrafine particles have stronger effects compared to larger particles because they can penetrate deeper into the respiratory system and reach the alveoli [21,24,25]. Generally, the toxic effect of PM_2.5_ on the respiratory system is determined by its chemical speciation and size [26,27]. In the present study, we found that the most abundant elements in the PM_2.5_ were Ca, K, Na, Fe, Al, Mg, Ti, Zn, and Pb, followed by Mn, Cu, V, Ba, Cr, As, Ni, and Cd. In addition to metals, the PM_2.5_ also appeared to have a higher content of water-soluble inorganic ions (SO_4_^2−^ and NO_3_^−^). Organic carbon accounted for approximately 44.09 μg per mg of the PM_2.5_ samples, which was 70% higher than the amount of elementary carbon. Organic carbon compounds including quinones and polycyclic aromatic hydrocarbons (PAHs) are known to have a strong correlation with redox reactivities [28]. In vitro studies have shown that high metal and carbon contents are directly associated with an increase in oxidative stress and pro-inflammatory response [21,27]. Besides the concentrations, the size of PM is important in determining its toxicological effects, which are associated with its potential to induce inflammatory injury, oxidative damage, and other biological effects [1]. In our study, we characterized the PM_2.5_ with dynamic light scattering (DLS) and found that the average size of our samples was approximately 0.7 μm. The smaller size of our PM_2.5_ samples may contribute to exposure-induced chronic pulmonary injury.

### 4.2. PM_2.5_ Activates the Phagosome Pathway

PM_2.5_ triggered cellular uptake by forming phagosomes [29]. Our previous study showed that PM_2.5_-induced oxidative stress triggers an increase in the number of autophagosomes in human lung epithelial A549 cells [24]. In the present study, KEGG pathway and STRING analysis showed that cluster of differentiation 36 (CD36), Class III β-tubulin (TUBB3), cytochrome b β chain (CYBB), and myeloperoxidase (MPO) were related to phagosomes in PM_2.5_-treated lung. Platelet glycoprotein 4 (CD36) serves as a ligand receptor of collagen types I and IV, oxidized low-density lipoprotein, and long-chain fatty acids [30,31]. PM_2.5_ may be recognized by CD36, triggering formation of the phagosomes, then activating NADPH oxidase to form mature phagosomes and eventually activating MPO to become phagolysosomes.

### 4.3. PM_2.5_ Alters Redox Homeostasis

It has been well demonstrated that PM_2.5_-mediated oxidative stress is mainly caused by an imbalance between production of ROS and antioxidant defense activity [8,32]. In our analyses, 22% of the proteins identified are involved in redox homeostasis. Antioxidant defenses prevent the generation of the most reactive form of ROS and subsequent oxidative damage [32]. Hydrogen peroxide oxidoreductase (MPO) acts by producing hypochlorous acid (HOCl) which is likely to contribute to the tissue damage caused by neutrophils at sites of inflammation [33]. CYBA is a major component of NAD(P)H oxidase, and the NAD(P)H oxidase system is considered to be the most important source of superoxide anions [34]. COQ6 is required for the biosynthesis of coenzyme Q10 (COQ10), which is a component of the mitochondrial electron transport chain and also acts as a lipophilic antioxidant to protect cells from ROS [35]. CYP2A5 and GPX2 have been identified to exert a cytoprotective response against oxidative damage [36]. In this study, we found that MPO, CYBB, and COQ6, which play roles in intracellular redox homeostasis, showed an especially prominent increase in expression, whereas CYP2A5 showed markedly decreased levels of expression.

### 4.4. Proteins Involved in PM_2.5_-Induced Pulmonary Inflammation and Fibrosis

Inflammation is considered to have a crucial function in PM_2.5_-induced chronic obstructive pulmonary diseases (COPD), asthma, and lung cancer [37,38,39]. Previous experimental studies have demonstrated that serum levels of TNF-α, IL-6, and lactate dehydrogenase, as well as bronchoalveolar lavage fluid (BALF) protein concentration and pulmonary infiltration of neutrophils and macrophages, were increased in PM_2.5_-treated mice [40,41]. PM_2.5_ triggered cellular uptake by forming phagosomes [29]. CD36 can recognize collagen types I and IV, oxidized low-density lipoprotein, and long-chain fatty acids as ligands [30,31]. BPI fold-containing family B member 1 (BPIFB1, also called LPLUNC1) is secreted from airway epithelial cells and is also present in airway submucosal glands and minor glands of the oral and nasal cavities, which may play an important role in innate immunity in the lung. In the present study, we identified at least six inflammation- and immune-signaling-related proteins. They are platelet glycoprotein 4, disintegrin and metalloproteinase domain-containing protein 17, collagen alpha-2(IV) chain, collagen alpha-1(IV) chain, and BPI fold-containing family B member 1 and other inflammation-associated proteins between PM_2.5_ and FA lungs. Collectively, these results indicate that PM_2.5_-induced immune signaling participates in the pulmonary injury.

Pulmonary fibrosis is well characterized as an expansion of fibroblasts or myofibroblasts. Abnormal accumulation of the ECM plays an important role in fibrosis [30,42]. In the present study, the proteomics analyses demonstrated that PM_2.5_ induced ECM–receptor interaction, phagosomes, and two fibrosis signaling pathways in the lungs of mice. These results agree with the finding that ECM-regulated genes are over-represented in lung fibroblasts [42]. Fibroblast growth factor-1 (FGF-1), the prototypic member of the FGF family of growth factors, displays antifibrotic functions, down-regulating collagen expression and antagonizing some profibrotic effects of TGF-β [43]. In addition, we found that the PI3K/Akt pathway was activated in the lungs of PM_2.5_-treated mice, which has been found to elicit a survival signal against multiple apoptotic insults [44]. PI3K, as an important regulator of cell growth and survival, plays a crucial role in response to oxidative stress [45]. A recent study demonstrated that PI3K/Akt signaling is involved in the pathogenesis of bleomycin-induced pulmonary fibrosis [46,47].

## 5. Conclusions

In summary, the results of the present study showed that PM_2.5_ exposure induces an inflammatory response and collagen deposition in PM_2.5_-treated lung. The results of TMT-based quantitative proteomics analyses identified a series of differentially expressed proteins related to redox homeostasis, immune response, and fibrotic response. Moreover, our findings revealed that the activation of ECM–receptor interaction, phagosomes, and PI3K–Akt signaling cascades is involved in PM_2.5_-induced pulmonary fibrosis. Taken together, these results will provide new insights into the toxic mechanisms underlying PM_2.5_-elicited lung injury. However, the functions of key differentially expressed proteins of interest need be further evaluated in follow-up work.

## Figures and Tables

**Figure 1 ijerph-16-00098-f001:**
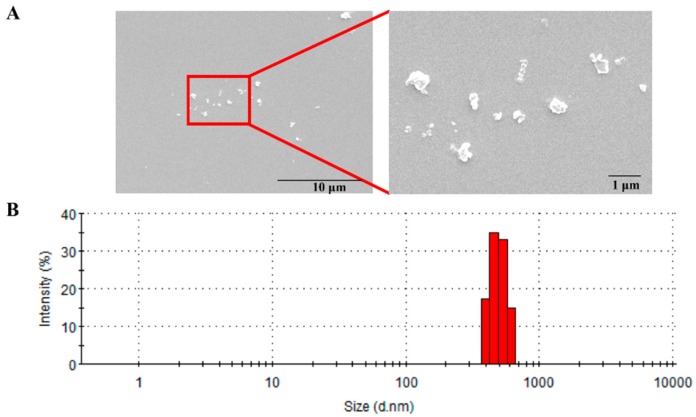
Particulate matter (PM_2.5)_ size distribution. (**A**) Scanning electron microscope (SEM) image of PM_2.5_ (magnification: 5000×–10,000×). (**B**) PM_2.5_ size distribution in ultra-pure deionized water was detected by dynamic light scattering (DLS).

**Figure 2 ijerph-16-00098-f002:**
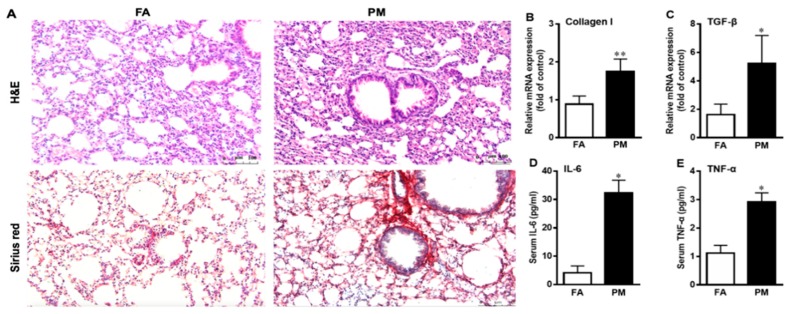
PM_2.5_ exposure induces pulmonary inflammation and fibrosis. (**A**) Lung sections from filtered air (FA) and particulate matter (PM_2.5_)-exposed mice were stained with hematoxylin and eosin (H&E) and Sirius red staining at 200× magnification. (**B**–**E**) After exposure for 12 weeks, the mRNA levels of fibrotic genes collagen I and transforming growth factor β (TGF-β) in lungs (**B**,**C**) and the level of inflammatory factors interleukin 6 (IL-6) and tumor necrosis factor α (TNF-α) in serum (**D**,**E**) were measured by qPCR and enzyme linked immunosorbent assay (ELISA) kits, respectively. Data are presented as mean ± standard deviation (SD); * *p* < 0.05, ** *p* < 0.01 vs. FA group.

**Figure 3 ijerph-16-00098-f003:**
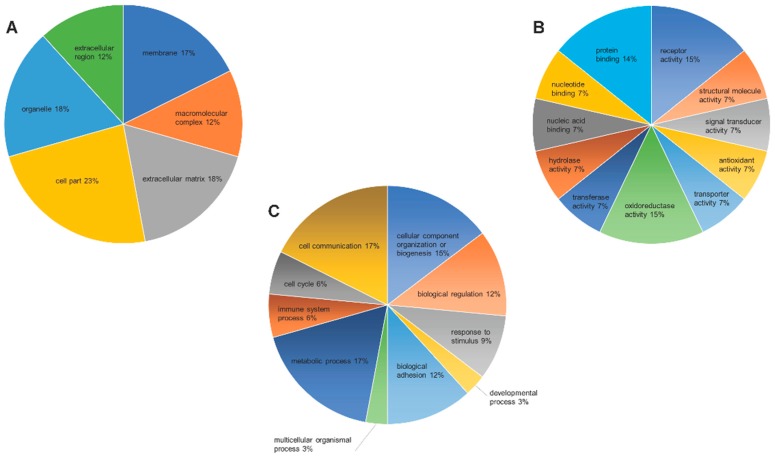
Gene Ontology (GO) analysis for differentially regulated proteins in the lung of PM_2.5_-exposed mice. The GO terms for cellular compartments (**A**), biological processes (**B**), and molecular functions (**C**) of the proteins are shown.

**Figure 4 ijerph-16-00098-f004:**
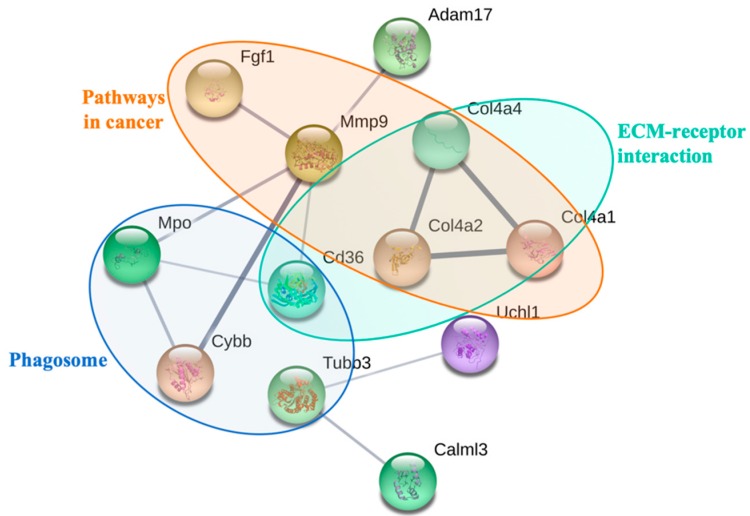
The analysis of the functional interaction network of PM_2.5_-regulated proteins by the search tool for the retrieval of interacting genes/proteins (STRING) algorithm.

**Figure 5 ijerph-16-00098-f005:**
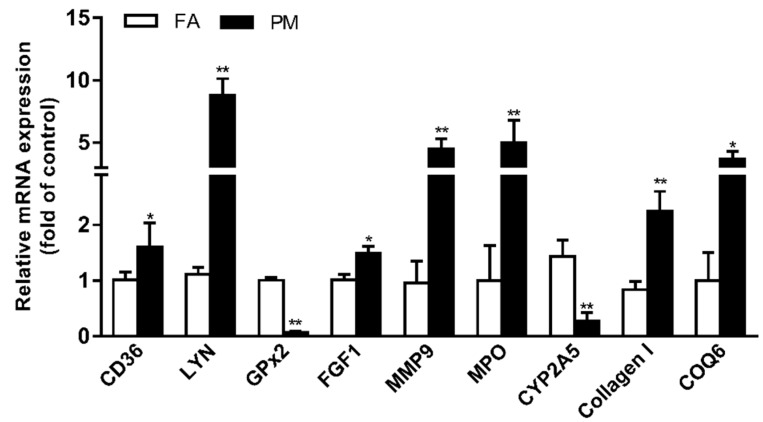
Real-time polymerase chain reaction analysis confirming the gene expression profiles obtained from proteomic data in the lungs of PM_2.5_-exposed mice. Data are presented as the mean ± SD of three independent experiments. * *p* < 0.05; ** *p* < 0.01—vs. the FA group. FA: filtered air, PM: particulate matter.

**Table 1 ijerph-16-00098-t001:** Chemical composition analysis of the PM_2.5_ samples.

	Component	Concentration (μg/mg)
Metal elements	Ca	13.76 ± 0.30
	K	10.33 ± 0.03
	Na	7.96 ± 0.08
	Fe	7.06 ± 0.13
	Al	6.24 ± 0.06
	Mg	3.98 ± 0.09
	Ti	1.66 ± 0.02
	Zn	1.62 ± 0.01
	Pb	0.54 ± 0.002
	Mn	0.35 ± 0.004
	Cu	0.20 ± 0.003
	V	0.13 ± 0.002
	Ba	0.13 ± 0.001
	Cr	0.081 ± 0.001
	As	0.064 ± 0.001
	Ni	0.045 ± 0.001
	Sr	0.045 ± 0.003
	Cd	0.014 ± 0.001
	Mo	0.11 ± 0.001
	Cs	0.006 ± 0.0003
	Co	0.004 ± 0.0008
Inorganic ions	SO_4_^2−^	212.71 ± 3.67
	NO_3_^−^	97.52 ± 6.81
	NH_4_^+^	85.61 ± 0.52
	Cl^−^	1.85 ± 0.41
Carbon	OC	44.09 ± 0.32
	EC	25.51 ± 0.75
	OC/EC	1.72 ± 0.65

**Table 2 ijerph-16-00098-t002:** Kyoto Encyclopedia of Genes and Genomes (KEGG) pathway analysis of the differentially expressed proteins involved in PM_2.5_-induced pulmonary injury.

Pathway	Term	Count	*p*-Value	Molecular Pathway
1	ECM–receptor interaction	4	0.0010	COL4A4 ↑, CD36 ↑, COL4A2 ↑, COL4A1 ↑
2	Pathways in cancer	6	0.0017	FZD1 ↑, COL4A4 ↑, COL4A2 ↑, FGF1 ↑, MMP9 ↑, COL4A1 ↑
3	Phagosome	4	0.0072	CYBB ↑, MPO ↑, TUBB3 ↑, CD36 ↑
4	Small cell lung cancer	3	0.0160	COL4A4 ↑, COL4A2 ↑, COL4A1 ↑
5	Protein digestion and absorption	3	0.0175	COL4A4 ↑, COL4A2 ↑, COL4A1 ↑
6	Amoebiasis	3	0.0297	COL4A4 ↑, COL4A2 ↑, COL4A1 ↑
7	PI3K–Akt signaling pathway	4	0.0458	COL4A4 ↑, COL4A2 ↑, FGF1 ↑, COL4A1 ↑

ECM: extracellular matrix.

**Table 3 ijerph-16-00098-t003:** Identified proteins from PM_2.5_-exposed lung.

Accession	Gene	PM/FARatio Average	Annotation
*Redox homeostasis*			
Q8R1S0	COQ6	1.619	Ubiquinone biosynthesis monooxygenase
P11247	MPO	1.391	Myeloperoxidase
Q61093	CYBB	1.306	Cytochrome b-245 heavy chain
Q3UBG2	PID1	1.298	PTB-containing, cubilin and LRP1-interacting protein
P20852	CYP2A5	0.584	Cytochrome P450 2A5
*Immune response*			
Q64524	H2BE	2.130	Histone H2B type 2-E
Q08857	CD36	1.428	Platelet glycoprotein 4
Q9Z0F8	ADAM17	1.302	Disintegrin and metalloproteinase domain-containing protein 17
P08122	COL4A2	1.319	Collagen alpha-2(IV) chain
P02463	COL4A1	1.313	Collagen alpha-1(IV) chain
Q61114	BPIFB1	0.531	BPI fold-containing family B member 1
*Fibrotic response*			
P61148	FGF1	1.558	Fibroblast growth factor 1
P41245	MMP9	1.434	Matrix metalloproteinase-9
Q08857	CD36	1.428	Platelet glycoprotein 4
Q9QZR9	COL4A4	1.362	Collagen alpha-4(IV) chain
P08122	COL4A2	1.319	Collagen alpha-2(IV) chain
P02463	COL4A1	1.313	Collagen alpha-1(IV) chain

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
