# Peer review of "TMT-Based Quantitative Proteomics Analysis Reveals Airborne PM2.5-Induced Pulmonary Fibrosis"

_ijerph, 2018, doi:10.3390/ijerph16010098_

Round 1

Reviewer 1 Report

An additional immunohistochemistry verification is needed to determine the levels of these proteins in the control and exposed lung tissue slides.  

Author Response

An additional immunohistochemistry verification is needed to determine the levels of these proteins in the control and exposed lung tissue slides. 

Response:Thank you very much for the kindly suggestion! It is indeed convincing that evaluating PM2.5 toxicity in lung injury by immunohistochemistry verification. We had done this experiment before, unfortunately, we did not achieve a good result. Therefore, we used qPCR to verify the differential proteins according to the methods by Wang, et al. and Kim et al. (BMC Plant Biology 2016; 16(1): 199; Toxicology and Applied Pharmacology 2018; 355: 28-42). We will carry out the immunohistochemistry verification in the future study. Thanks a lot!

Reviewer 2 Report

Thank you very much for the opportunity to review this paper that investigates airborne PM2.5-Induced pulmonary fibrosis in animal models. The study is from China and overall I recommend the publication of this manuscript. I do have some minor comments which I have mentioned below. Also, some sections of the paper need some editing by a native English speaker. 

Lines 28-31: This sentence seems to end abruptly and is too long as well. Consider rewriting or rephrasing it. 

Line 49: Delete 'are' from the sentence. 

Line 92: It would be better if the authors can describe in short this procedure rather than expecting the reader to search for Wan et al. paper online. 

Lines 304-305: Rewrite this sentence please. ...'potential impacts on toxicological effects'....It is better to say that the PM2.5 speciation helps in the understanding of the toxicological effects...or something on those lines. 

Lines 311-314: This is a long sentence again. Please rewrite it or divide it into two shorter sentences as the preset sentence seems to end abruptly. 

Line 322: Please add 'were' before form....

Line 359: The conclusion is just too small. Please highlight the main findings etc. of your work in the conclusion and any updates on future work that would add on to the already existing body of literature. 

General comment: It would be nice to see the findings from this study being compared with similar such studies conducted in China or other parts of the world. It will help accentuate the importance of the research work done by this group. 

Author Response

Lines 28-31: This sentence seems to end abruptly and is too long as well. Consider rewriting or rephrasing it. 

Response: Thank you very much for your suggestions! We have corrected these sentence, please check Lines 28-30.

Line 49: Delete 'are' from the sentence.

Response: Thank you for your kindly comments, we deleted it, please check Line 49. 

Line 92: It would be better if the authors can describe in short this procedure rather than expecting the reader to search for Wan et al. paper online. 

Response: Thank you for your suggestions. We corrected it, please check Line 87.

Lines 304-305:Rewrite this sentence please. ...'potential impacts on toxicological effects'....It is better to say that the PM2.5 speciation helps in the understanding of the toxicological effects...or something on those lines. 

Response: Thanks a lot! We rewrite the sentence, please check Lines 304-306.

Lines 311-314: This is a long sentence again. Please rewrite it or divide it into two shorter sentences as the preset sentence seems to end abruptly.

Response: Thank you for the critique. We rewrite the this sentence, please check Lines 312-314. 

Line 322: Please add 'were' before form....

Response: Thank you for your suggestion! We add 'were' in Line 324.

Line 359: The conclusion is just too small. Please highlight the main findings etc. of your work in the conclusion and any updates on future work that would add on to the already existing body of literature. 

Response: Thank you very much for your suggestions! We revised the conclusion section, please check Lines 363-370.

Reviewer 3 Report

Please make sure that you cite the appropriate literature references that best support the statements made in your paper.

The materials and methods sections appear to be thorough and well described.

You present a lot of very detailed and important information in your results sections with appropriate use of tables and figures. The flow of the text discussion could benefit from some editing to improve clarity.

I would suggest that you consider editing the Discussion section to separate specific topics and improve the flow. As currently written it is a bit difficult to follow, particularly the discussion of how the findings from your study fits into the large body of literature for potential molecular mechanisms of fine particulate related injury to the lungs.    

Author Response

Please make sure that you cite the appropriate literature references that best support the statements made in your paper.

Response: Thank you very much for your kind suggestions. We checked and added some literatures into our paper, including Line 47 References [2,4,5], Line 322 References [8,31] and Line 361 References [45,46].

The materials and methods sections appear to be thorough and well described.

Response: Thank you very much for your comment!

You present a lot of very detailed and important information in your results sections with appropriate use of tables and figures. The flow of the text discussion could benefit from some editing to improve clarity.

I would suggest that you consider editing the Discussion section to separate specific topics and improve the flow. As currently written it is a bit difficult to follow, particularly the discussion of how the findings from your study fits into the large body of literature for potential molecular mechanisms of fine particulate related injury to the lungs.    

Response: Thank you very much for your suggestions! We revised the Discussion section. The subtitles were added to separate specific topics (Please check Line 303, 311, 320 and 334). 
